# *CDH1* Mutation Distribution and Type Suggests Genetic Differences between the Etiology of Orofacial Clefting and Gastric Cancer

**DOI:** 10.3390/genes11040391

**Published:** 2020-04-03

**Authors:** Arthavan Selvanathan, Cheng Yee Nixon, Ying Zhu, Luigi Scietti, Federico Forneris, Lina M. Moreno Uribe, Andrew C. Lidral, Peter A. Jezewski, John B. Mulliken, Jeffrey C. Murray, Michael F. Buckley, Timothy C. Cox, Tony Roscioli

**Affiliations:** 1New South Wales Health Pathology, Prince of Wales Hospital, Randwick, Sydney 2031, Australia; arthavan.selvanathan@health.nsw.gov.au (A.S.); ying.zhu@health.nsw.gov.au (Y.Z.); michael.buckley@health.nsw.gov.au (M.F.B.); 2Discipline of Child and Adolescent Health, University of Sydney, Sydney 2031, Australia; 3Canterbury Health Laboratories, Canterbury District Health Board, Christchurch 8011, New Zealand; cynixon.15@gmail.com; 4Genetics of Learning Disability Service, Waratah, Newcastle 2298, Australia; 5The Armenise-Harvard Laboratory of Structural Biology, Department of Biology and Biotechnology, University of Pavia, 27100 Pavia, Italy; luigi.scietti@unipv.it (L.S.); federico.forneris@unipv.it (F.F.); 6Department of Orthodontics & the Iowa Institute for Oral and Craniofacial Research, University of Iowa, Iowa, IA 52242, USA; lina-moreno@uiowa.edu; 7Lidral Orthodontics, Rockford, MI 49341, USA; acl_iowa@yahoo.com; 8Institute of Oral Health Research, School of Dentistry, University of Alabama at Birmingham, Birmingham, AL 35294, USA; pjetski1010@icloud.com; 9Department of Plastic and Oral Surgery, Boston Children’s Hospital, Boston, MA 02215, USA; john.mulliken@childrens.harvard.edu; 10Harvard Medical School, Boston, MA 02115, USA; 11Department of Pediatrics, University of Iowa, Iowa, IA 52242, USA; jeff-murray@uiowa.edu; 12Departments of Oral & Craniofacial Sciences, School of Dentistry, and Pediatrics, School of Medicine, University of Missouri-Kansas City, Kansas City, MO 64108, USA; 13Centre for Clinical Genetics, Sydney Children’s Hospital - Randwick, Sydney 2031, Australia; 14Prince of Wales Clinical School, University of New South Wales, Sydney 2031, Australia; 15NeuRA, University of New South Wales, Kensington, Sydney 2031, Australia

**Keywords:** orofacial clefting, cleft lip, cleft palate, gastric cancer, cadherin 1, genotype-phenotype correlation

## Abstract

Pathogenic variants in *CDH1*, encoding epithelial cadherin (E-cadherin), have been implicated in hereditary diffuse gastric cancer (HDGC), lobular breast cancer, and both syndromic and non-syndromic cleft lip/palate (CL/P). Despite the large number of *CDH1* mutations described, the nature of the phenotypic consequence of such mutations is currently not able to be predicted, creating significant challenges for genetic counselling. This study collates the phenotype and molecular data for available *CDH1* variants that have been classified, using the American College of Medical Genetics and Genomics criteria, as at least ‘likely pathogenic’, and correlates their molecular and structural characteristics to phenotype. We demonstrate that *CDH1* variant type and location differ between HDGC and CL/P, and that there is clustering of CL/P variants within linker regions between the extracellular domains of the cadherin protein. While these differences do not provide for exact prediction of the phenotype for a given mutation, they may contribute to more accurate assessments of risk for HDGC or CL/P for individuals with specific *CDH1* variants.

## 1. Introduction

E-cadherin (epithelial cadherin) is the archetypical member of the classic cadherin family of calcium-dependent cell-adhesion molecules. Encoded by the *CDH1* gene, E-cadherin is the principle adhesive protein of epithelial adherens junctions, playing a major role in both tissue morphogenesis and epithelial differentiation. The extracellular region of mature E-cadherin comprises five extracellular (EC) domains that mediate adhesion. Strong and stable adhesion requires chelation of Ca^2+^ ions in each linker region that separates the EC domains as well as coordination of corresponding cytoskeletal changes mediated through its cytoplasmic tail. Mechanistically, calcium binding stabilizes EC domain flexibility and exposes an N-terminal tryptophan (Trp) residue, which embeds in a pocket of the EC1 domain on a Cadherin protomer that is *in trans*. Cell–cell adhesion is then consolidated and strengthened by cumulative *cis* interactions between *trans*-dimerized proteins [1,2,3].

Over the last decade, the role of E-cadherin in cancer etiology has been intensively investigated, identifying that pathogenic variants in *CDH1* are present in multiple types of cancer, including hereditary diffuse gastric cancer (HDGC) and lobular breast cancer [4,5]. More recently, both germline and *de novo* pathogenic variants in *CDH1* have also been shown to underlie both syndromic (blepharocheilodontic syndrome; BCDS) [6,7] and non-syndromic forms of cleft lip with or without cleft palate (CL/P) [8,9,10].

Functional studies on cancer-associated, as well as a limited number of CL/P-associated, E-cadherin missense variants have identified varying degrees of impact on cell–cell adhesion. A variety of mechanisms, including reduced *trans*-dimerization, increased endocytic recycling, and loss of cytoskeletal interaction and subsequent signal transduction, have been found to explain this impact [10,11,12,13,14,15]. Despite this, CL/P has only been reported in a few families with *CDH1*-linked HDGC and likewise HDGC has only been infrequently reported in families with *CDH1*-linked CL/P.

Frebourg et al. (2006) reported the first two families in which individuals presented with both HDGC and CL/P. They described two families with splice site variants that resulted, at least in lymphocytes, in complex aberrant splicing that included one transcript predicted to produce a protein with an in-frame deletion [16]. Based on this, they hypothesized that such variants may have a trans-dominant negative impact, distinguishing them from other variants. However, variants affecting these canonical splice sites causing in-frame deletions have been reported subsequently in families with purely HDGC [17,18] or purely CL/P; hence it is unclear whether such a hypothesis of a dominant negative effect holds true in all cases. 

Figueiredo et al. (2019) reviewed available literature from 1985 to 2018 on *CDH1* germline variants but did not identify preferential type or location of *CDH1* variants that would help direct differential patient management [4]. Obermair et al. (2019) noted that families with a combined phenotype of HDGC and CL/P had variants within the extracellular domains (ECD) [19]; however, overlap was noted for families with isolated HDGC. The clinical relevance of differentiating craniofacial from cancer phenotypes is substantial, given the possibility of identifying *CDH1* variants in genomic investigations for CL/P. Likewise, the identification of *CDH1* variants in familial HDGC raises challenges for counselling couples on the additional risk of having a child with CL/P. 

In this study, we have undertaken a review of available molecular data from published and unpublished reports over the past 20 years of patients with HDGC and CL/P. Our study differs from other recent reviews in a number of important ways, including the restriction of our assessment to variants classified using current ACMG criteria as at least ‘likely pathogenic’ (i.e., exclusion of variants of uncertain significance (VUSs)), and analysis of the location of missense variants on the three-dimensional protein structure rather than the two-dimensional linear structure. From this analysis, and in contrast to prior studies, we note different characteristics between the variants in the two distinct clinical presentations, including in variant type, and their location in the protein. In particular, we note a strong preponderance for CL/P-related pathogenic variants to lie around the linker regions between extracellular domains, where the chelation of calcium ions occurs to stabilize the extracellular structure of E-cadherin and promote strong trans-cellular adhesion. These observations could contribute towards developing an algorithm to enable characterization of the phenotype from the genotype and, at minimum, lead to improved risk assessment for genetic counselling of patients. We further propose alternative or complementary mechanisms to explain the dichotomous clinical impact of *CDH1* mutations that provide future opportunities for investigation.

## 2. Materials and Methods 

### 2.1. Literature Searches

To generate a comprehensive list of all previously reported pathogenic variants in *CDH1*, a PubMed search for articles from 2000 to 2019 involving *CDH1* and any of ‘cleft lip/palate’, ‘hereditary diffuse gastric cancer/HDGC’ or ‘blepharocheilodontic syndrome/BCDS’ was undertaken. Articles were reviewed for strictly germline variants reported to be associated with HDGC, CL/P, or BCDS which were then collated [5,6,8,9,10,12,16,17,18,19,20,21,22,23,24,25,26,27,28,29,30,31,32,33,34,35,36,37,38,39,40,41,42,43,44,45,46,47,48,49,50,51,52]. Local sequencing results from a cohort including five previously unreported patients with CL/P (unpublished data) were also included. A further search was conducted of the Leiden Open Variation Database (LOVD) [25] and the ClinVar Database [17]. Variants were accepted for inclusion from these databases if they had appropriate phenotypes and met ACMG criteria for ‘Likely Pathogenic’ or ‘Pathogenic’ (see Appendix A, Appendix A, for full list of variants and their classification). The *CDH1*-specific modified ACMG criteria created by the ClinGen expert panel were also considered in the assessment of HDGC variants [53]. Articles reporting somatic mutations within tumors were excluded. Variants were grouped into three categories based on phenotype: ‘HDGC’, ‘HDGC+CL/P’, and ‘CL/P’.

### 2.2. Characterization of Type of Mutation

Within each phenotype group, variants were categorized according to their type: missense variants, in-frame deletions, ‘start codon lost’ variants, truncating variants (including nonsense variants, frameshift variants, partial and entire exon deletions), and splice region variants. The in-frame deletions were grouped with missense mutations for statistical analysis, and the ‘start codon lost’ and truncating variants were combined, because of similarities in their predicted effect on the protein. Differences between the phenotype groups were assessed using the Chi-squared test.

### 2.3. Characterization of Mutation Location

Exonic mutations were grouped, based on data from the UniProt database [54], as falling within the signal/pro-peptide region (S/PP), extracellular region (ER), or ‘transmembrane and intracellular’ (TM/IC) region. The proportions of mutations located in these three regions were compared across phenotype groups using Fisher’s exact test. The proportions of missense variants in each group that occurred in the ‘linker region’ between extracellular domains were mapped on the mouse E-cadherin ectodomain three-dimensional structure (PDB 3Q2V) [3] using PyMol [55], and compared using the Chi-squared test.

### 2.4. Characterization of Missense Mutations by In Silico Prediction Scores

Once compiled, mutations were entered into the VarCards database [56]. Each variant was assessed using up to 23 different predictive in silico tools, providing a deleterious:all (D:A) algorithm score. The Combined Annotation Dependent Depletion (CADD) score was included separately given its utility in assessing pathogenicity of non-missense variants. Variants were uploaded in appropriate format to the CADD database [57]; the output data were created using CADD v1.4. Variants from the ‘HDGC’, ‘HDGC+CL/P’, and ‘CL/P’ groups were compared in regard to their pathogenicity scores using the Kruskal–Wallis Test [58].

### 2.5. Characterization of Missense Mutations by Amino Acid Tolerance

Tolerance of individual amino acids to substitution was assessed using MetaDome, a platform assessing tolerance to variation developed at the Centre for Molecular and Biomolecular Informatics at the Radboud University Medical Centre in Nijmegen [59]. This database assesses each amino acid change by the ratio of non-synonymous/synonymous changes (dn/ds) at homologous protein domains throughout the genome. There were insufficient data in the ‘HDGC+CL/P’ group for statistical analysis; hence the remaining two groups (‘HDGC’ and ‘CL/P’) were compared using the Mann–Whitney U Test. 

## 3. Results

### 3.1. Differences in Mutation Type Between Phenotype Groups

Collating all of the available literature (see Appendix A), there were a total of 280 variants in *CDH1* analyzed. This comprised 245 (88%) mutations causing HDGC, 27 (10%) mutations causing CL/P, and eight (3%) manifesting as both phenotypes within the same pedigree. The mutations were grouped into nonsense, missense, and splice mutations as in Table 1.

Nonsense mutations comprised 71% of mutations in the ‘HDGC’ group but represented only 38% and 11% of variants in the ‘HDGC+CL/P’ and ‘CL/P’ groups, respectively, consistent with an upward trend in the presence of a cancer phenotype. Conversely, there was a marked preponderance of missense mutations in the ‘CL/P’ group (70%) compared with the ‘HDGC’ group (4%). The ‘HDGC+CL/P’ group had a high proportion of splice variants (50%) compared to the other two groups (24% and 19%, respectively). Given there were comparatively few variants in all classes of the ‘HDGC+CL/P’ group, we removed this group from the statistical analysis. There was a statistically significant difference in the proportions of different types of mutations between the ‘HDGC’ and ‘CL/P’ groups (Chi-square statistic 109.5, *p* < 0.00001). 

### 3.2. Differences in Location of Variants Between Phenotype Groups

The location of missense variants, in-frame deletions, start codon lost variants, and coding region truncating variants were represented on the E-cadherin protein primary sequence (Figure 1) while splice variants were represented on the *CDH1* gene structure (Figure 2). These representations appeared to show a number of differences in the distribution of different variant types between the ‘HDGC’ and ‘CL/P’ groups. Whilst the numbers of variants are small in two of the groups, we note that all splice variants (5/5) in the ‘CL/P’ group reside at the same splice donor site (exon 9-intron 9 boundary). None of those splice variants in the ‘HDGC’ (0/59) and ‘HDGC+CL/P’ (0/4) groups are located at this donor site; they are spread over other donor and acceptor sites or create new sites. Splice variants at this junction generally lead to in-frame deletions involving parts of the third extracellular domain, and hence are thought to have a deleterious effect on protein function. However, this notable difference between the ‘HDGC’ and ‘CL/P’ groups indicates that there could be differential impacts of splice variants in *CDH1* that warrant further experimental follow up.

For non-truncating (missense and in-frame) variants, where there are sufficient numbers for statistical comparison, we also observed a difference in distribution. To quantify this, we divided variants into those that occurred in the signal/pro-peptide (amino acids 1–154), extracellular (155–697), and transmembrane/intracellular (amino acids 698–882) regions. Table 2 demonstrates that 18% of ‘missense and in-frame’ HDGC variants occur within the signal/pro-peptide region, with 73% in the extracellular region, and 9% in the transmembrane/intracellular region. This is in contrast to CL/P variants, where no ‘likely pathogenic’ or ‘pathogenic’ variants have been reported in the signal/pro-peptide region, and 89% occur within the extracellular region. In head-to-head comparison between the ‘HDGC’ and ‘CL/P’ groups, this difference was not statistically significant (Fisher’s Exact Test *p* = 0.16). The one missense variant in the ‘HDGC+CL/P’ group occurred within the extracellular region but was not included in the statistical analysis due to limited sample numbers.

### 3.3. Localization of CL/P Variants to Linker Regions 

Figure 2 provides a linear representation of the locations of variants stratified by group. Despite there being no statistically significant difference by region on a broad scale, CL/P variants appeared to be more frequently located at or near the linker regions between individual extracellular domains than HDGC variants. Given the importance of this area in calcium-binding and providing stability to the overall extracellular region, we pursued this further by mapping CL/P and HDGC missense variants (and in-frame deletions) onto a three-dimensional E-cadherin protein structure (Figure 3). In contrast to the linear mapping where seven of the CL/P variants appeared to map in the linker regions, this 3D mapping demonstrated that 13 of the 19 missense variants cluster around linker regions, with an additional variant involving the key tryptophan (W156) facilitating strong *in trans* interaction. A further two of the six remaining variants (V412A and T522I) not within the defined linker regions mapped to positions immediately adjacent the defined regions. The interaction of distinct residues to form the 3D linker regions is shown on the linear structure depicted in Figure 4. This figure also demonstrates the stark contrast with the variants that cause HDGC: none of the 10 missense variants, nor the in-frame deletion variant, occur within the linker regions (although L583R and the two distinct F626V variants map immediately adjacent to the defined regions). This difference is statistically significant (Chi-square >10.5, *p* < 0.001). The single missense variant seen in the ‘HDGC+CL/P’ group also maps to the linker region. Of note, two HDGC missense variants (D244G and I326N) map to the *cis*-interface—the interacting surfaces between EC1 and EC2 from two independent *in cis* E-cadherin monomers [3].

### 3.4. Lack of Differences in In Silico Prediction Scores, and Amino Acid Tolerance to Missense Substitution, Between Phenotype Groups

The median D:A proportions for variants were 0.82 in the ‘HDGC’ group, 0.78 in the ‘HDGC+CL/P’ group, and 0.91 in the ‘CL/P’ group. The median CADD scores for the same three groups were 30.0, 29.8, and 26.5. Neither of these in silico score differences were statistically different using the Kruskal–Wallis test (*p* = 0.31 and *p* = 0.42, respectively). This is consistent with variants reaching a threshold for pathogenicity in all *CDH1*-associated phenotypes but that in silico models are unable to further differentiate whether a variant is likely to result in HDGC or CL/P. Similarly, the median MetaDome dn/ds score for the missense variants from the three groups were 0.61, 0.19 (only one sample), and 0.52, respectively, with no statistically significant difference between the ‘HDGC’ and ‘CL/P’ groups (U = 99, *p* = 0.48, figure not shown). 

## 4. Discussion

*CDH1* encodes E-cadherin, a major transmembrane adhesion protein of epithelial adherens junctions. Mature, plasma membrane-localized E-cadherin is composed of five extracellular domains (EC1-EC5), as well as a transmembrane and a short intracytoplasmic domain that facilitates connection to both the microtubule and actin cytoskeletons [60]. The adhesive activity of E-cadherin is controlled by both the levels of membrane localized protein and as well by the extracellular concentration of calcium ions. Extracellular calcium, when chelated by the linker regions between each EC domain, effectively rigidifies the extracellular domains, promoting stronger adhesion between E-cadherin protein on adjacent cells. Epithelial cell–cell adhesion is dynamically regulated by extracellular cues, including both calcium and various growth factors, as well as by intracellular signaling events [61]. 

Underpinning the importance of tight regulation of adhesion in epithelial cells, adhesion strength is inversely correlated with the proliferative capacity of epithelia. Adhesion must therefore be sufficient to maintain the protective function of an intact epithelium yet permit growth of the epithelial layer as needed [62]. The ability to readily modulate the cell–cell adhesive strength also determines the behavior of entire epithelial tissues that characterize key morphogenetic events during embryogenesis, defined by tissue fusion in palatogenesis, epithelial-to-mesenchymal transitions in neural crest cell formation, and branching morphogenesis that underpins glandular development [60]. Loss, or deregulation, of intercellular adhesion is also characteristic of epithelial-derived tumors and their metastases [62].

Genetic linkage analysis and subsequent DNA sequencing has identified germline *CDH1* mutations as a primary cause of HDGC (70%–80% lifetime risk with a positive family history) and lobular breast cancer in women (40% lifetime risk) [63]. However, germline *CDH1* mutations have also been identified in individuals with both sporadic and familial forms of cleft lip/palate. In such cases, the CL/P can be syndromic (blepharocheilodontic syndrome; BCDS) or non-syndromic in its presentation [5,8,9,46,48]. Surprisingly, *CDH1* mutations contribute to presentation of both cancer and CL/P in 3% of families. The nature of how germline mutations in *CDH1* lead to such broadly different phenotypes remains under investigation. 

### 4.1. Evidence for Genotype–Phenotype Correlation for CDH1 Mutations

There is evolving evidence in the literature regarding which *CDH1* mutations are associated with HDGC compared with CL/P. We have identified a statistically significant difference in the frequency of missense versus nonsense mutations in HDGC compared with CL/P in the reported cohort of patients with *CDH1* mutations. Although it is not yet possible to automatically predict whether a *CDH1* mutation is HDGC-causing or CL/P-causing, extensive bioinformatic and functional characterization of the effects of mutations may ultimately facilitate development of a robust algorithm to assist in predicting which phenotype is more likely. Phenotypic variability is a common feature of many diseases and this variation is at least in part due to differences in the type of pathogenic variant [64]. For example, most classically, protein-truncating or frameshift mutations in the *DMD* gene cause the X-linked Duchenne Muscular Dystrophy; missense mutations or in-frame deletions generally cause the milder Becker Muscular Dystrophy [65].

It has been noted previously by Obermair et al. (2019) that *CDH1* variants in families with both CL/P and HDGC are found in the extracellular domains [19]. Our review of the literature supports this, although there are only eight pathogenic/likely pathogenic variants reported in the context of both phenotypes. We noted a similar preponderance within the CL/P group. This observation cannot however be used in isolation to predict phenotype, given there are also cases of HDGC caused by mutations in this region (and, in a few cases, the same residue). 

Building on the work of Obermair et al. (2019) [19], our analysis of *CDH1* variants demonstrated that those encoding the linker regions between the extracellular domains are ‘hot spots’ for causing CL/P. This includes an area of the protein that is highly intolerant to missense substitution (dn/ds <0.25) from amino acids 253–260, which is responsible for chelation of calcium ions between the first and second EC domains. Five missense variants associated with CL/P occur in this region, as compared to none of the likely pathogenic/pathogenic HDGC variants considered in this study. It should be noted that variants in this region have been identified in patients with HDGC. These were not included in this review either because they were identified as somatic variants only or they did not currently meet the ACMG criteria for classification as likely pathogenic or pathogenic. Another eleven variants (three in-frame deletions, eight missense variants) in CL/P also occur close to the three-dimensional space occupied by calcium-binding sites between EC domains. 

Despite the differences identified from this analysis of all reported pathogenic and likely pathogenic *CDH1* variants in HDGC and CL/P, none—either in isolation or collectively—accurately predicts which phenotype a variant will cause. There are likely to be additional mutational mechanisms underlying the dichotomous phenotypes, and these are discussed below. 

### 4.2. Other Potential Mutational Mechanisms in Genes Encoding Multiple Phenotypes 

Observing mutational mechanisms in other genes may shed light on further potential explanations for the phenotypic spectrum of germline *CDH1* mutations. Disorders such as spinal muscular atrophy have phenotypic variability due to modifier genes [66]. It is conceivable that a modifier gene, acting as part of an oligogenic model, reduces the severity of (or even prevents) cleft/lip palate in some patients, or reduces risk of gastric cancer. Genome-wide association studies certainly support such an oligogenic or threshold model, and in such cases many of the additional ‘influencing loci’ contain likely regulatory variants that affect expression of genes *in cis*. Another mechanism underlying phenotypic variability is altered splicing efficiency. Mutations that reduce the length of the polythymidine sequence of intron 8 (IVS8) in *CFTR* reduce efficiency of exon 9 splicing in patients with a R117H/C mutation. Poor splicing efficiency leads to a more severe phenotype (cystic fibrosis), whereas patients with improved splicing have a milder phenotype (isolated congenital absence of the vas deferens) [67]. Such a mechanism has been suggested in one family with HDGC by Zhang et al. (2014) [42]. Here, one member of the family—the only one who did not present with HDGC—carried the same pathogenic *CDH1* variant as other affected family members but, in addition, also carried a common neighboring splice variant that none of the other family members inherited. Functional studies of patients with additional splice site variants in *CDH1*, as well as a broader search for intronic variants via whole genome sequencing, may be beneficial in characterizing this as a potential mechanism. 

Given variability in penetrance and severity of presentation of CL/P is also seen in many inbred mouse models, other factors also need to be considered. Foremost among such possible factors are epigenetic contributions that variably impact gene expression. Altered methylation, histone modification and imprinting are all mechanisms that would cause phenotypic variability with the same genetic change. Prader–Willi syndrome and Angelman syndrome are a key example of parental imprinting (via methylation) causing vastly different phenotypes. Studies of more extensive pedigrees would help evaluate whether this mechanism is contributing to *CDH1* pleiotropy.

Finally, many types of cancer are believed to arise as a result of “two mutational hits”—a principle germline mutation and a subsequent somatic mutation [68]. These ‘second’ hits typically arise in somatic tissue and can be in a different gene or in the second allele of the same gene [15,69]. In many cases of HDGC, a second, somatic hit in *CDH1* (frequently affecting promoter methylation or less often, loss of heterozygosity) has been described [70,71] and thus may further reduce the adhesive strength below a threshold in that specific tissue, resulting in deregulated growth [72]. Alternatively, a second hit in a cell cycle regulator may increase the proliferative potential of cells already harboring reduced E-cadherin adhesive activity, promoting a similar outcome [69]. One must also consider then the possible contribution of somatically-arising variants or epigenetic differences during embryogenesis as potential contributors to CL/P penetrance and variability. The embryonic facial prominences that form the lip and palate are some of the most rapidly dividing tissues and, at least conceptually, even a moderate impact somatic mutation or change in gene expression as a result of methylation differences could sufficiently affect the growth rate and hence disrupt the critical timing of fusion of already compromised early facial tissues. In support of this hypothesis, recent modeling of neural crest cell migration into the developing chick face based on live cell imaging data has suggested that even a fairly moderate reduction in the rate of cell division (< 20%) ultimately results in insufficient neural crest cells reaching their final destination in the anterior region of the developing face [73]. Such an impact on growth prior to fusion could be a major risk factor for cleft presentation. 

## 5. Conclusions

This study provides evidence for some differences in *CDH1* germline mutation type and location that are involved in creating the encountered phenotypic heterogeneity of CL/P and HDGC. In particular, we have identified that variants lying in close proximity to the ‘linker regions’ are more likely to be associated with CL/P. However, these differences are not yet robust enough to reliably differentiate these phenotypes prospectively. Databases of *CDH1* variants and their clinical consequences will therefore have a substantial impact on genetic counselling in people where pathogenic *CDH1* variants are detected. There may be other factors which are yet to be elucidated, such as modifier genes, altered splicing efficiency, epigenetic phenomena, or somatic mutations, that could be important contributory factors in defining susceptibility or risk of each condition. It is hoped that identification and analysis of further families may help unveil the mechanism underlying this pleiotropy, which in turn could lead to more specific clinical management.

## Figures and Tables

**Figure 1 genes-11-00391-f001:**
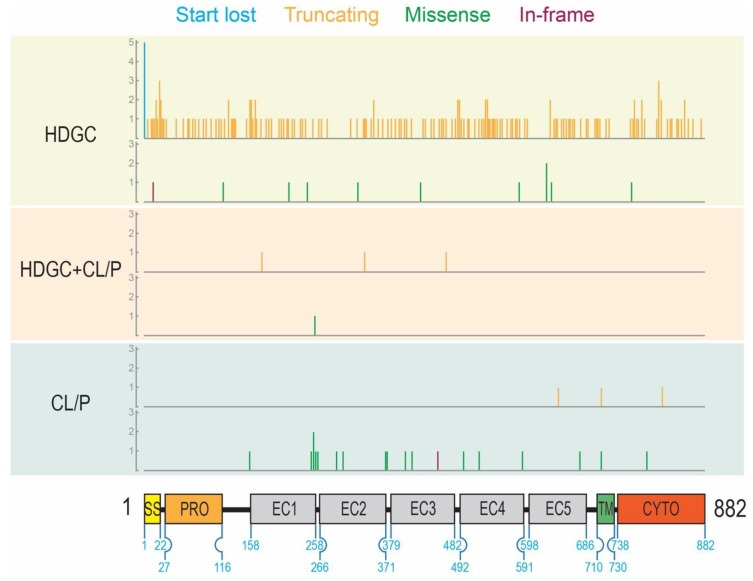
Location of all variants with respect to the epithelial cadherin (E-cadherin) domain structure grouped by mutation type and phenotype (HDGC: hereditary diffuse gastric cancer; CL/P: cleft lip/palate). Variant positions are marked by vertical lines that correspond on the x-axis to the amino acid residue number. The height of the vertical lines corresponds to the number of variants located at that given residue position (y-axis). The color of the vertical lines represents the type of variant: start lost (blue), truncating (orange), missense (green), and in-frame deletion (purple). For each phenotype, variants are grouped by type: start lost and truncating variant (upper); missense and in-frame deletion variants (lower). For reference, numbers on the schematic of the protein represent the start and end residues of each domain.

**Figure 2 genes-11-00391-f002:**
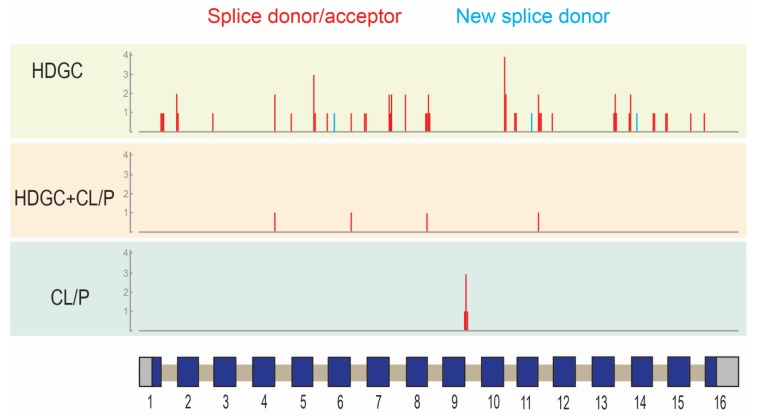
Location of splice variants in *CDH1* grouped by phenotype (HDGC: hereditary diffuse gastric cancer; CL/P: cleft lip/palate). Vertical lines correspond to the approximate location of the variants with respect to the gene structure (schematic; x-axis). Exons are shown as boxes: coding region (blue); untranslated regions (grey). Introns (tan) and exons are not drawn to scale. The height of the vertical lines corresponds to the number of variants located at that given position (y-axis). The color of the vertical lines represents the type of variant: splice donor/acceptor variants (red), new splice donor (light blue).

**Figure 3 genes-11-00391-f003:**
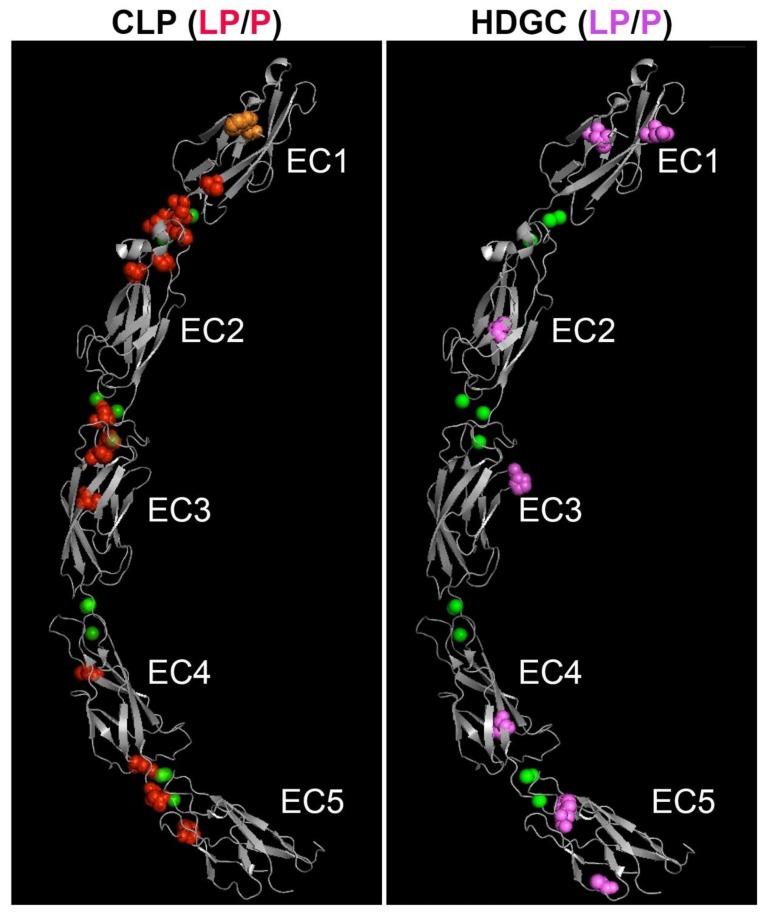
Homologous location of likely pathogenic and pathogenic human missense and in-frame deletion variants causing CL/P (left panel) and HDGC (right panel) on the three-dimensional ectodomain structure of E-cadherin. The extracellular region of mature mouse E-cadherin (PDB 3Q2V) [3], comprised of five EC domains, is shown in grey. The positions of CL/P variants are shown as red spheres (left image); HDGC variants are shown as pink spheres (right image). The location of the CL/P variant in the tryptophan (W156) that is critical for *in trans* interaction of E-cadherin is shown in orange. Chelated calcium ions are shown as green spheres. Note apparent clustering of CL/P variants around the linker regions, in contrast to the HDGC variants. LP/P—likely pathogenic/pathogenic.

**Figure 4 genes-11-00391-f004:**
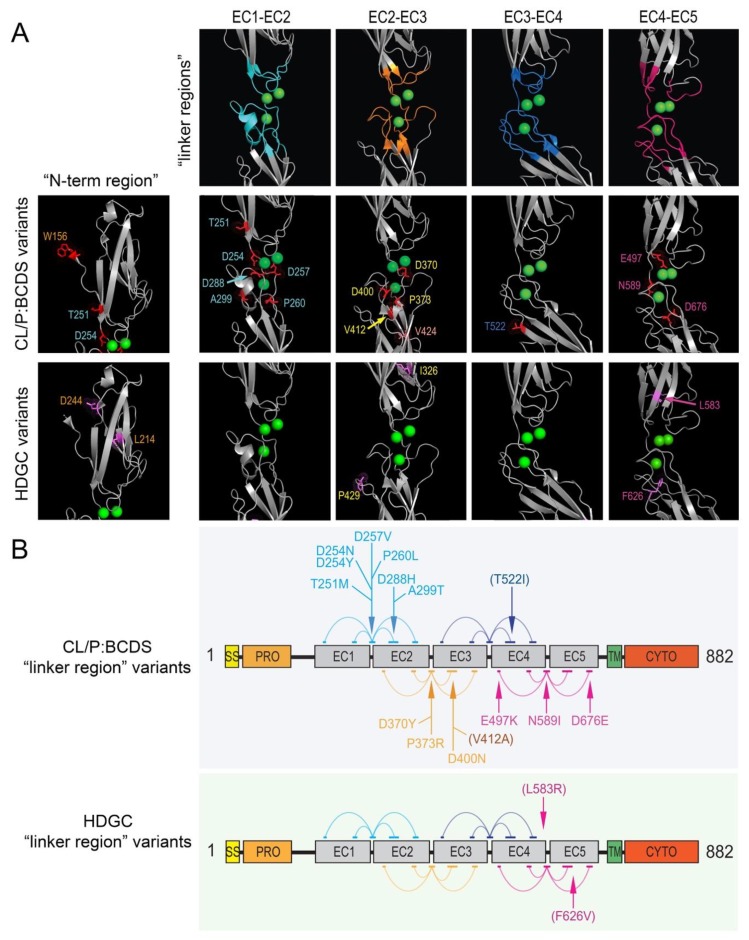
Most pathogenic/likely pathogenic CL/P missense variants, but very few HDGC variants, cluster in the linker regions between the extracellular domains of E-cadherin. (**A**) Residues contributing to each linker are colored on the 3D structure (upper panel in (**A**)) in teal (EC1-EC2 linker), orange (EC2-EC3 linker), blue (EC3-EC4 linker), and pink (EC4-EC5 linker). The locations of pathogenic/likely pathogenic missense CL/P variants (middle panel) and missense HDGC variants (lower panel) that map to near the respective 3D linker region structures are shown (colored and labeled). (**B**) Schematic of the E-cadherin primary sequence showing the different domains (rectangles): SS—single sequence; PRO—pro domain; EC1–5—extracellular domains 1–5; TM—transmembrane domain; CYTO—cytoplasmic (intracellular) domain. A total of 15 of the 17 CL/P missense variants found within the extracellular region (and 15 of 19 total) are part of, or one residue adjacent to, the linker regions marked by arrows, as compared to only three (L583R and both F626V) of the 11 HDGC variants. Note: five distinct clusters of amino acids (horizontal colored lines joined by arcs in the schematic), spread across the primary sequence of two adjacent EC domains (grey rectangles) contribute to each of the respective linker regions in the 3D protein structure. As in (**A**), the clusters of amino acid residues in the schematic are colored in teal, orange, blue, and pink for those contributing to the EC1-EC2 linker, EC2-EC3 linker, EC3-EC4 linker, and EC4-EC5 linker, respectively. The variants indicated in brackets in (**B**), reside immediately adjacent a defined cluster.

**Table 1 genes-11-00391-t001:** Numbers of variants by variant type and phenotype.

	Nonsense	Missense	Splice	Total
**HDGC**	175	11	59	245
**HDGC+CL/P**	3	1	4	8
**CL/P**	3	19	5	27
**Total**	181	31	68	280

HDGC: hereditary diffuse gastric cancer; CL/P: cleft lip/palate.

**Table 2 genes-11-00391-t002:** Missense and in-frame variants in each phenotypic group by location.

	S/PP	ER	TM/IC	Total
**HDGC**	2	8	1	11
**HDGC+CL/P**	0	1	0	1
**CL/P**	0	17	2	19
**Total**	2	26	3	31

S/PP: signal/pro-peptide region; ER: extracellular region; TM/IC: transmembrane and intracellular region.

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
