# Peer review of "CDH1 Mutation Distribution and Type Suggests Genetic Differences between the Etiology of Orofacial Clefting and Gastric Cancer"

_genes, 2020, doi:10.3390/genes11040391_

Round 1
Reviewer 1 Report
The authors present a thorough analysis of an interesting problem, the fact that E-cadherin mutations are present in two very different diseases (Hereditary diffuse gastric cancer and a hereditary form of Cleft Lip and palate). There has been little clinical overlap between the two groups, but the types of mutations that lead to one versus the other are unclear. While other review articles have approached this same question recently, this paper both adds additional unreported CL/P patients and also limits the mutations analyzed to ones that are predicted to be pathogenic. It adds additional evidence to the importance of the linker regions, though it fails to provide any data on how these might be important, leaving that to the field for additional studies. It also highlights the importance of missense vs truncating mutations.
In general, the figures are well designed and informative.
In Figure 3, it would be helpful to add an additional panel which shows both sets of mutations, as it appears that the W156 region also has changes in HDGC, which would be interesting.
In Figure 4, it is unclear which mutations belong to which disease. These panels are difficult to interpret as they stand and the figure legends are unclear.
Author Response
Reviewer 1
We have taken on board the comments of Reviewer 1 and have added a panel showing both sets of variants into Figure 4. We have signposted more clearly which variants belong to which disease (by way of a descriptor beside the panel on the left), as well as by making the figure legend clearer.
Reviewer 2 Report
"CDH1 mutation distribution and type suggests genetic differences between the etiology of orofacial cleating and gastric cancer" is a very interesting manuscript. The quality of the manuscript is below average in terms of organization and preparation. Authors have prepared a very well structured article.Author Response
Reviewer 2
Thank you for taking the time to review our article and for your comments.
Reviewer 3 Report
The manuscript entitled "CDH1 Mutation Distribution and Type Suggests Genetic Differences Between the Etiology of Orofacial Clefting and Gastric Cancer" presents a nice piece of work with potencial utility, if improved.
I have some major concerns that I describe below, but many more are described in the attached marked PDF of the manuscript:
1- The overall reference selection is poor and does not mention seminal work from the field. This should be completely reformulated, as the message of the paper is many times incorrect. I left some suggestions along the manuscript, but more reference changes are needed.
2- The ACMG classification of variants, which is a major issue for the selection of presented variants is not well described nor presented. Detailed information and the rational for variant selection and classification should be presented in extra columns of the supplementary table.
3- There are many concepts that are not correct along the text and I left notes to all of them throughout the text.
4- The hypothesis advanced to differentiate the consequences between different CDH1 variant types are poor and disregard the very first publication that presents compelling evidence from human embryos and the expression levels of CDH1 across embryonic development.
5- I would urge authors to reply to all the comments that I have added along the text.

Author Response
Reviewer 3
Thank you for your substantial review and comments. We will approach these sequentially.
Line 57 and Line 60: In the early paragraphs we have updated the references accordingly to include the broader reviews suggested.
Lines 67-74:
REVIEWER COMMENT: "In this part of the introduction, it would be important to present the hypothesis that Frebourg et al, 2006 advanced for the occurrence of CL/P in CDH1 mutation carriers, that is related to the insufficient levels of E-cadherin arising from a single allele during specific stages of the embryonic development. Also an hypothesis raised in the presence of transcripts with in-frame exon skipping that would produce almost full length proteins with potential dominant negative effect.”
RESPONSE: We agree with the importance of the Frebourg et al. (2006) paper, as the initial paper that reported families with both HDGC and CL/P. Hence, we have introduced a separate paragraph regarding it in the main text (now lines 67-73). However, while the hypothesis put forward by Frebourg et al. provides one plausible contributing mechanism and explanation, there are some limitations. Further, their hypothesis is based on findings regarding splice site mutations in just two families, and does not take into account all the splicing consequences they report. Nor does it sufficiently explain the variety of germline splicing variants, or the variation in position and functional impact of germline missense variants that have been subsequently reported.
Germline mutations in CDH1, whether splice mutations, missense changes, truncating/nonsense mutations or complete deletion of the allele, effectively all reduce adhesion strength and/or impact the dynamics of adhesion regulation in all epithelial tissues that normally express E-cadherin, both during development and in adult tissues. Like Frebourg et al, we agree that the facial ectoderm is more sensitive than many other ectodermal/epithelial regions of the developing embryo, principally because of the remarkably quick changes in epithelial adhesion strength that are thought to mediate the ectodermal morphological changes that occur during lip development to mediate tissue fusion. Based on the finding of splice site mutations in only two families with both HDGC and CL/P, Frebourg et al proposed a trans-dominant negative effect of the CL/P-associated splice-mutations. This hypothesis was premised on the observation that both of their reported splice mutations give rise to an alternatively splice mRNA predicted to result in a protein with an in-frame deletion. While this alternate form of E-cadherin may indeed have a trans-dominant impact, just as many other E-cadherin mutations are predicted to have, they also reported that multiple other splice forms also arose as a result of the specific mutation in each family - these other splice forms predicted to result in premature truncation and NMD. Unfortunately, no data was presented in their paper to quantify the relative levels of each mRNA splice form. Further, these splicing effects were assessed in peripheral blood lymphocytes from the patients, a cell type that is not of epithelial origin and that expresses very limited amounts of CDH1 mRNA. While it is reasonable to propose that such aberrant splicing occurs in all cell types, the differential expression of splicing regulators in different cell types means that the relative impact of any given mutation in splicing may vary between cell types. Further, the impact of the one splice form with an in-frame deletion is unknown: if representing 5% of aberrant transcripts, or even 50%, what impact is sufficient to result in a clinical phenotype? Mice, for example, show no signs of cancer or CL/P when the levels of E-cadherin are reduced by 50%. Furthermore, Frebourg et al also inferred, based on an analysis of COGENE data (microarray-based analysis of gene expression in tissues from human embryos), that CDH1 expression is appropriately but dynamically expressed in different parts of the face and in tissues consistent with its role in CL/P. While this is true for the mRNA, levels of E-cadherin protein, and indeed cell surface amounts of the protein, are also separately and tightly regulated, making it difficult to draw conclusions about the impact of one splice isoform when multiple are present. One also needs to consider that other CDH1 splice mutations (some affecting the same exon-intron boundary, and some affecting others) have been reported in HDGC families without CL/P. This includes variants at exon-intron boundaries of an exon that, if skipped, would produce in-frame deletions. We have mentioned this in the text accordingly.
Lines 100-101:
REVIEWER COMMENT: The supplementary table that compiles all the collected data should be mentioned here. Also that supplemental table should be enriched with a column to report whether variants are pathogenic or likely pathogenic based the ACMG criteria and another with the same information for the LOVD database. Another problem with the supplemental table is the lack of standardization for nomenclature of CDH1 variants. All variants should have both coding sequence-related and protein-related nomenclature.
RESPONSE: We have included a reference to the supplementary table at this point, and have also enriched that table with the ACMG classification of all variants to demonstrate that they are likely pathogenic or pathogenic. Three variants (c.367C>T, c.1320G>A c.531-18G>A) had been classified as pathogenic or likely pathogenic based on literature, but we have now re-classified these three as VUS after carefully reviewing and strictly applying the ACMG criteria: although we suspect that this classification will change once functional evidence is obtained, for now these were removed from the dataset. As mentioned, the full list of CDH1 variants was pulled from ClinVar and LOVD as well as supplemented with new variants from our own cohort. As expected, more variants are reported in ClinVar than in the LOVD, and thus we cannot provide LOVD classification for a majority of the variants listed and so have not included this in the table. That said, the LOVD classification is more strict than that applied in ClinVar and there was no discrepancy between the LOVD and ACMG classification for the variants that were pulled from this database. With respect to the nomenclature, we believe the nomenclature of the variants is already standardized in the table: variants that affect splicing do not have protein-related nomenclature (the examples of the impact of splice variants shown in Frebourg et al, 2006 highlight the challenges in doing so because of the often multiple splice isoforms that may result). Furthermore, each splice variant has been listed consistently relative to the c.DNA position (and genomic coordinate as per convention) to demonstrate their proximity to nearby splice variants. If a predicted splice variant lies within an exon, then we also show the predicted impact on the amino acid, shown in brackets.
Line 134:
REVIEWER COMMENT: Refer to the suppl table where this information should be visible. These 286 variants, are not actually 286 different variants. Indeed, what authors are referring here in the number of probands carrying these CDH1 variants. This should be completely clarified, as may are recurrent variants.
RESPONSE: We have referred to the supplementary table as suggested with thanks. The number of variants was reduced to 283 as discussed above, and the statistics have been altered accordingly. However, contrary to the reviewer’s impression, each variant in the table is actually distinct: although some may indeed show the same amino acid change, the nucleotide changes are different, hence its separate representation. Indeed, there are variants that are seen in multiple probands (ie. hotspots) but these are NOT listed more than once.
Line 138:
REVIEWER COMMENT: Please provide the ClinVar classification for these variants. If the premise is that in the current study all variants have been revised and only likely pathogenic or pathogenic variants are under study, this classification should be available for the readers. One of these five variants (c.532-18G>A) is benign and should not be considered. The variant "c.1320+5G>A" is not reported in Clinvar, but a similar one c.1320+5G>C is reported a single time and is classified as VUS. Please clarify the classification of this variant for the purpose of this study. Have all these variants been classified as Likely pathogenic or Pathogenic? How many different variants are included here? How many specific variants are included in different clinical groups?
RESPONSE: We did not feel it relevant to provide the ClinVar classification for each individual variant given many variants are misclassified in ClinVar. We therefore elected to instead, as suggested, provide AMCG classification for each variant. We agree regarding the variant c.532-18G>A being classified as a VUS, and hence have removed it from the dataset using the formal criteria. The variant c.1320+5G>A is reported by Kievit et al. (2018) [PMID: 29348693] as likely pathogenic. It is mentioned as a de novo change (PS2) that also has poor predictive scores (PP3) and is absent from controls (PM2). We have therefore elected to leave this variant in the dataset. Please see Table 1 for the number of variants in each phenotypic group.
Lines 156-157:
REVIEWER COMMENT: Indeed, both exons 8 and 9 of the CDH1 gene are in-frame exons and given this fact, variants that affect their exon-intron splicing junctions more often give rise to transcripts with in-frame exon skipping than transcripts arising from cryptic splicing and premature termination codons leading to NMD. So conceptually this sentence is not correct.
RESPONSE: We agree with this criticism and have modified this sentence accordingly. The splice variants affect a functional domain and so are still classified as ‘PVS1_Strong’ using the modifications of the PVS1 criterion as per Abou Tayoun et al. (2018).
Line 161:
REVIEWER COMMENT: In order to have real comparison for this group of variants, it would be important to have them classified as bona fide "pathogenic" or "likely pathogenic" variants. A quick look into the supplemental material was sufficient to understand that many of these variants are either classified as VUS, conflicting or do not even have an entry in ClinVar. Also, something that is not clear is whether being classified as "pathogenic" or "likely pathogenic" means the same for HDGC and for CL/P. Wider studies are needed to clarify whether a variant that mildly predisposing for HDGC can indeed predispose to CL/P. Collected numbers and phenotypes are not yet sufficient to disclose this. However, under this rationale, the classification of CDH1 variants in the context of HDGC and CL/P should indeed be different, because variants that are VUS, LB or B for HDGC may be causal for CL/P. For comparison purposes, I agree that authors should agree to a type of classification, however the hypothesis of conditional causality depending on the clinical phenotype should be discussed.
RESPONSE: The use of ACMG criteria for classification is one of the accepted means within the genetics community of classifying variants in terms of pathogenicity. That said, it is acknowledged that no current classification system is perfect as each is based on meeting (in any combination of ways) different criteria from a collection of observations (clinical, bioinformatic, functional, animal model), each with their own inherent limitations. In this study, we chose to utilize the ACMG criteria and guidelines, independently applied to each variant irrespective of their prior classification using another system or claim in a publication.
With regard to the criteria for classification, the reviewer raises a very interesting point about whether ACMG classification is independent of disease phenotype. We believe that classification is independent of disease phenotype overall, though there are certain criteria (PP4 and PS2 for instance) which do look at the presence (and extent) of phenotypic overlap. The purpose of this study, however, is to investigate if there are features about variants that might give insight in the basis of the different phenotypes resulting from CDH1 variants and thus form the basis of a more accurate classification system to distinguish them, with obvious impact for clinical management and counselling. Nevertheless, this issue should be re-evaluated as classification systems are improved or modified.
Line 264:
REVIEWER COMMENT: This reference is absolutely not adequate. Authors should find a reference that provides reliably this information, preferably coming from original papers or from high profile review articles about CDH1 and HDGC.
RESPONSE: We believe this is a very fair criticism by the reviewer, and so have modified the reference to the Pharoah et al. paper (2001) instead, which is a high-profile review article.
Lines 268-269:
REVIEWER COMMENT: The paper that identified for the first time a correlation between CDH1 germline variants in HDGC and CL/P (Frebourg et al 2006) presents an hypothesis that should be herein discussed.
RESPONSE: The hypothesis of the Frebourg et al. (2006) paper has been discussed above and in the text (lines 67-73).
Lines 293-296:
REVIEWER COMMENT: This should be bullet proof. Indeed, the supplementary table provided is very poor in demonstrating how each variant was classified according to ACMG, or the exact ClinVar classification. This is mandatory for this manuscript that uses the ACMG classification to justify genotype-phenotype correlations.
RESPONSE: We believe we have addressed this criticism by the modification of the Supplementary Digital Content (Table 1).
Lines 330-334:
REVIEWER COMMENT: This information is not accurate and the original papers that thoroughly described CDH1 second hits in HDGC are not mentioned. In those papers, it is evident that the most frequent second hit in HDGC tumors is promoter methylation followed by Loss of heterozygosity and not a second mutation.
RESPONSE: We have modified that sentence accordingly to mention promoter methylation and loss of heterozygosity as additional key mechanisms in the second hit hypothesis.
Lines 336-341:
REVIEWER COMMENT: This is a rather difficult hypothesis to accept. Authors should indeed go back to the paper that pioneered the correlation between CL/P and CDH1 and look at the data from human embryos that provides a rationale for the lack of sufficient E-cadherin available in a heterozygous individual at key points during development, namely when the midline is closing close to the upper lip and palate.
RESPONSE: We agree that one contributing factor as to why CL/P is seen with CDH1 mutations versus broader ectodermal phenotypes is that this region of the developing face either exhibits differential expression levels or the cellular changes required for lip morphogenesis, or both require more precise regulation of adherens junction strength. Developmentally, however, CDH1 mRNA expression levels are not the best correlate of E-cadherin protein levels on the surface or within adherens junctions, as there is considerable post-translation regulation (eg. endocytosis just for one example) that is now recognized as a major means of controlling adhesion strength and morphological change of epithelia. Also, the analysis done by Frebourg et al, was an analysis of COGENE data (no longer available publicly) which is microarray-based mRNA expression data, which is notoriously not the most robust as a quantitative method. Furthermore, the Frebourg et al paper, while appropriate to recognize as the first example of CDH1 variants in familial cases with both CL/P and HDGC, based their hypothesis on their two cases, which by most standards would be deemed insufficient as evidence of a correlation. Taken together, we do not believe that simple differences in mRNA expression levels are sufficient to explain the incidence or variable penetrance and expressivity of clefting, especially when considering the variability within families including monozygotic twins and inbred mice. It also does not clearly explain why a germline loss of function allele in CDH1 causes HDGC in multi-affected multigenerational families without any appearance of clefting, while other similar loss of function alleles are inherited in multi-affected multigenerational families with only CL/P and no HDGC.
Our proposition that somatic mutations could also contribute to the phenotypic presentation of disorders such as CL/P is based on recent whole genome sequencing data from numerous studies that show more than an order of magnitude higher rate of mutation in somatic tissues when compared to germline rate of mutation – including not only single nucleotide somatic variation but also marked structural variation within all tissues. In proposing this as a possible contributing mechanism, we certainly are not suggesting it is the principal mechanism nor necessarily a major mechanism - just that is should be considered as a possible mechanism. To help explain the rationale for this hypothesis, we have added some additional explanation as well as now citing recently-published work to provide conceptual support for such a model. As the reviewer points out with respect to HDGC, differential methylation and other epigenetic factors are also potential (or likely) contributing factors, as recognized for many other developmental disorders.
Round 2
Reviewer 3 Report
The manuscript improved to some extent, however there are still a number of items that need further work.
Most are related to the information that is present in the Supplementary table, which I detail below.
Lines 100-101:
REVIEWER COMMENT: The supplementary table that compiles all the collected data should be mentioned here. Also that supplemental table should be enriched with a column to report whether variants are pathogenic or likely pathogenic based the ACMG criteria and another with the same information for the LOVD database. Another problem with the supplemental table is the lack of standardization for nomenclature of CDH1 variants. All variants should have both coding sequence-related and protein-related nomenclature.
RESPONSE: We have included a reference to the supplementary table at this point, and have also enriched that table with the ACMG classification of all variants to demonstrate that they are likely pathogenic or pathogenic. Three variants (c.367C>T, c.1320G>A c.531-18G>A) had been classified as pathogenic or likely pathogenic based on literature, but we have now re-classified these three as VUS after carefully reviewing and strictly applying the ACMG criteria: although we suspect that this classification will change once functional evidence is obtained, for now these were removed from the dataset. As mentioned, the full list of CDH1 variants was pulled from ClinVar and LOVD as well as supplemented with new variants from our own cohort. As expected, more variants are reported in ClinVar than in the LOVD, and thus we cannot provide LOVD classification for a majority of the variants listed and so have not included this in the table. That said, the LOVD classification is more strict than that applied in ClinVar and there was no discrepancy between the LOVD and ACMG classification for the variants that were pulled from this database. With respect to the nomenclature, we believe the nomenclature of the variants is already standardized in the table: variants that affect splicing do not have protein-related nomenclature (the examples of the impact of splice variants shown in Frebourg et al, 2006 highlight the challenges in doing so because of the often multiple splice isoforms that may result). Furthermore, each splice variant has been listed consistently relative to the c.DNA position (and genomic coordinate as per convention) to demonstrate their proximity to nearby splice variants. If a predicted splice variant lies within an exon, then we also show the predicted impact on the amino acid, shown in brackets.
Reviewer Response: Regarding the supplementary table, there are still some fundamental problems:
- The classification in the table should comply with the information from the paper DOI:1002/humu.23650“Specifications of the ACMG/AMP Variant Curation Guidelines for the Analysis of Germline CDH1 Sequence Variants”.
- The analysis of the first 10 lines of the table immediately showed a great degree of inconsistency in variant classification. Please see the table attached with my comments in red in an extra column and in several places of the table. The same type of analysis should be performed for all variants. Indeed, the ClinVar ID for each variant should be added to support the classification presented in the paper.
- There are variants that are repeated, and I now understand it is related with different nucleotide changes. Given the lack of the nucleotide change information in the table, they will look like repetitions with a few mistakes to less informed readers.
- I disagree with the authors about the variant nomenclature. Indeed for the same protein variation, several nucleotide changes may occur, sometimes with different impacts. So indeed the mRNA nomenclature should be stated in the table. Please see the additional column that I have added.
- My intention with requesting more clarity about the classification of variants is related to the use of the scientific literature to actually classify variants for clinical purposes. Authors should acknowledge the fact that the classification of a variant as Likely pathogenic or Pathogenic determines whether a person is submitted to surgical removal of stomachs and breasts in an asymptomatic state. So, the responsibility associated with publishing accurate classifications is extremely high. This is important for clinical and fundamental research papers.
Line 134:
REVIEWER COMMENT: Refer to the suppl table where this information should be visible. These 286 variants, are not actually 286 different variants. Indeed, what authors are referring here in the number of probands carrying these CDH1 variants. This should be completely clarified, as may are recurrent variants.
RESPONSE: We have referred to the supplementary table as suggested with thanks. The number of variants was reduced to 283 as discussed above, and the statistics have been altered accordingly. However, contrary to the reviewer’s impression, each variant in the table is actually distinct: although some may indeed show the same amino acid change, the nucleotide changes are different, hence its separate representation. Indeed, there are variants that are seen in multiple probands (ie. hotspots) but these are NOT listed more than once.
Reviewer Response: Now it is clear, and it reinforces my comment above “I disagree with the authors about the variant nomenclature. Indeed, for the same protein variation, several nucleotide changes may occur, sometimes with different impacts. So indeed, the mRNA nomenclature should be stated in the table. Please see the additional column that I have added.”
This comment from the author reveals the need to add the nucleotide change and the reference transcript to the table.
Line 138:
REVIEWER COMMENT: Please provide the ClinVar classification for these variants. If the premise is that in the current study all variants have been revised and only likely pathogenic or pathogenic variants are under study, this classification should be available for the readers. One of these five variants (c.532-18G>A) is benign and should not be considered. The variant "c.1320+5G>A" is not reported in Clinvar, but a similar one c.1320+5G>C is reported a single time and is classified as VUS. Please clarify the classification of this variant for the purpose of this study. Have all these variants been classified as Likely pathogenic or Pathogenic? How many different variants are included here? How many specific variants are included in different clinical groups?
RESPONSE: We did not feel it relevant to provide the ClinVar classification for each individual variant given many variants are misclassified in ClinVar. We therefore elected to instead, as suggested, provide AMCG classification for each variant. We agree regarding the variant c.532-18G>A being classified as a VUS, and hence have removed it from the dataset using the formal criteria. The variant c.1320+5G>A is reported by Kievit et al. (2018) [PMID: 29348693] as likely pathogenic. It is mentioned as a de novo change (PS2) that also has poor predictive scores (PP3) and is absent from controls (PM2). We have therefore elected to leave this variant in the dataset. Please see Table 1 for the number of variants in each phenotypic group.
Reviewer Response: The information in Clinvar is not perfect, but is better curated that in any other database. For instance, for the variant W409R. This variant has been classified as likely benign by the curation team from ClinGen which is clearly described in the paper “Specifications of the ACMG/AMP Variant Curation Guidelines for the Analysis of Germline CDH1 Sequence Variants” and related with the frequency of a variant in carriers with or without phenotype of HDGC.
If authors used the ACMG criteria to classify variants, the available evidence should be available. Bear in mind that ACMG criteria for CDH1 are different from the general ones.
Line 161:
REVIEWER COMMENT: In order to have real comparison for this group of variants, it would be important to have them classified as bona fide "pathogenic" or "likely pathogenic" variants. A quick look into the supplemental material was sufficient to understand that many of these variants are either classified as VUS, conflicting or do not even have an entry in ClinVar. Also, something that is not clear is whether being classified as "pathogenic" or "likely pathogenic" means the same for HDGC and for CL/P. Wider studies are needed to clarify whether a variant that mildly predisposing for HDGC can indeed predispose to CL/P. Collected numbers and phenotypes are not yet sufficient to disclose this. However, under this rationale, the classification of CDH1 variants in the context of HDGC and CL/P should indeed be different, because variants that are VUS, LB or B for HDGC may be causal for CL/P. For comparison purposes, I agree that authors should agree to a type of classification, however the hypothesis of conditional causality depending on the clinical phenotype should be discussed.
RESPONSE: The use of ACMG criteria for classification is one of the accepted means within the genetics community of classifying variants in terms of pathogenicity. That said, it is acknowledged that no current classification system is perfect as each is based on meeting (in any combination of ways) different criteria from a collection of observations (clinical, bioinformatic, functional, animal model), each with their own inherent limitations. In this study, we chose to utilize the ACMG criteria and guidelines, independently applied to each variant irrespective of their prior classification using another system or claim in a publication.
With regard to the criteria for classification, the reviewer raises a very interesting point about whether ACMG classification is independent of disease phenotype. We believe that classification is independent of disease phenotype overall, though there are certain criteria (PP4 and PS2 for instance) which do look at the presence (and extent) of phenotypic overlap. The purpose of this study, however, is to investigate if there are features about variants that might give insight in the basis of the different phenotypes resulting from CDH1 variants and thus form the basis of a more accurate classification system to distinguish them, with obvious impact for clinical management and counselling. Nevertheless, this issue should be re-evaluated as classification systems are improved or modified.
Reviewer Response: I agree that no classification system is perfect, and I diverge from authors about the importance of genotype-phenotype consequences. Indeed different genetic mechanisms affecting the same gene can cause different phenotypes, and this is known for many diseases. I believe that CDH1 will not be an exception.
If authors wish to improve the CDH1 classification, the proposed one need to be based on the most accurate criteria established by the authors as the most reliable.
Line 264:
REVIEWER COMMENT: This reference is absolutely not adequate. Authors should find a reference that provides reliably this information, preferably coming from original papers or from high profile review articles about CDH1and HDGC.
RESPONSE: We believe this is a very fair criticism by the reviewer, and so have modified the reference to the Pharoah et al. paper (2001) instead, which is a high-profile review article.
Reviewer Response: The reference added misses the title of the paper.
Lines 293-296:
REVIEWER COMMENT: This should be bullet proof. Indeed, the supplementary table provided is very poor in demonstrating how each variant was classified according to ACMG, or the exact ClinVar classification. This is mandatory for this manuscript that uses the ACMG classification to justify genotype-phenotype correlations.
RESPONSE: We believe we have addressed this criticism by the modification of the Supplementary Digital Content (Table 1).
Reviewer Response: I would suggest further modifications, please see attached table file
Lines 330-334:
REVIEWER COMMENT: This information is not accurate and the original papers that thoroughly described CDH1 second hits in HDGC are not mentioned. In those papers, it is evident that the most frequent second hit in HDGC tumors is promoter methylation followed by Loss of heterozygosity and not a second mutation.
RESPONSE: We have modified that sentence accordingly to mention promoter methylation and loss of heterozygosity as additional key mechanisms in the second hit hypothesis.
Reviewer Response: This is not sufficient. There are at least two papers that need to be mentioned here:
PMID: 10973239 DOI: 10.1038/79120
PMID: 19269290 DOI: 10.1053/j.gastro.2009.02.065
